# Global Ban on Plastic and What Next? Are Consumers Ready to Replace Plastic with the Second-Generation Bioplastic? Results of the Snowball Sample Consumer Research in China, Western and Eastern Europe, North America and Brazil

**DOI:** 10.3390/ijerph192113970

**Published:** 2022-10-27

**Authors:** Ewa Kochanska, Katarzyna Wozniak, Agnieszka Nowaczyk, Patrícia J. Piedade, Marilena Lino de Almeida Lavorato, Alexandre Marcelo Almeida, Ana Rita C. Morais, Rafal M. Lukasik

**Affiliations:** 1Research and Innovation Center Pro-Akademia, 9/11 Innowacyjna Str., 95-050 Konstantynow Lodzki, Poland; 2Unidade de Bioenergia e Biorrefinerias, Laboratório Nacional de Energia e Geologia I.P., Estrada do Paço do Lumiar 22, 1649-038 Lisboa, Portugal; 3BISA Benchmarking Inteligência Sustentável, Av. Nove de Julho—Bela Vista, São Paulo CEP 01313-001, Brazil; 4Commitment Prestação Serviços, São Paulo CEP 05844-120, Brazil; 5School of Engineering, The University of Kansas, 4165E Learned Hall, Lawrence, KS 66045, USA

**Keywords:** bioplastic, food waste, consumer acceptance, snowball sampling

## Abstract

Plastic can be used for many things and at the same time is the most versatile material in our modern world. However, the uncontrolled and unprecedented use of plastic comes to its end. The global ban on plastic brings significant changes in technology but even more so in civil perception—changes taking place before our eyes. The aim of this study was to find answers to the questions about the readiness of consumers for a global ban on plastic. Within the research, the differences in consumer acceptance in countries in Europe, South and North America and Asia and the expression of social readiness to change attitudes towards plastic food packaging were analyzed. This work sketches the legal framework related to limiting the use of one-use food packaging made of fossil raw materials at the level of the European Union, Poland and Portugal but also at the level of the two largest economies in the world, China and the United States, as well as lower-income countries, e.g., Ukraine and Brazil. The survey results were analyzed using descriptive and inferential statistics. The performed study demonstrates that, in in all the surveyed countries, appropriate legal acts related to the reduction of plastic in everyday life are already in place. Furthermore, this work demonstrates the full understanding of plastic banning in all surveyed countries. Consumers are aware that every effort should be made to prevent the world from drowning in plastic waste. Society is, in general, open to the use of bioplastics produced from the second-generation resource if second-generation bioplastics contribute to environmental and pollution reduction targets.

## 1. Introduction

As a global society, we are drowning in plastic garbage. There is so much of it in our cities, forests, seas and even in the air that the United Nations has declared a state of a plastic disaster on Earth [1]. The General Assembly of the United Nations asks the dramatic question: “Planet or Plastics?” and shows the rising numbers of the plastics catastrophe: 500,000,000,000 plastic bags are used each year, 13,000,000 tons of plastic leak into the ocean each year, 17,000,000 barrels of oil are used for plastic production each year, 1,000,000 plastic bottles are bought every minute and 50% of consumer plastics are single-use [1]. According to Atlas of Plastic [2], the planet could reach more than 600 million tons of plastic produced annually by 2025. The vast majority of it ends up as waste. Therefore, there are significant efforts made to find environmentally friendly materials that can replace the crude-oil-origin plastics but are, at the same time, as functional and cheap as classical ones. Yet, the main challenge towards this goal is to develop the final environmentally friendly product and to produce it from raw materials other than cereals or potatoes, which should primarily be dedicated to food or feed uses. The circular economy concept refers to an economy that uses a systems-focused approach and involves industrial processes and economic activities that are restorative or regenerative by design and aim to keep products, components and materials at their highest utility and value at all times [3]. Therefore, according to the circular economy concept, the waste is at the center of the intensive research for a technology dedicated to bioplastics production.

Due to the fact that the largest user of plastic is the packaging sector, and food packaging accounts for about 40% of it [4], it is crucial to understand the needs of this sector in bioplastic development. Although bioplastic has been known as an environmentally friendly material for more than 70 years [5], still, the definition from a technical and material point of view is neither simple nor unambiguous. The term of “bioplastics” is used interchangeably with “biodegradable polymers”. Bioplastics are defined as materials produced from raw materials other than fossil fuels, especially crude oil, coal and lignite, and which are degraded naturally with the assistance of microorganisms in aerobic or anaerobic processes [6]. According to the European Bioplastics association [7], from a technical point of view, bioplastics can be divided into three main groups, as sketched in Figure 1:
Bio-based or partially bio-based but non-biodegradable materials such as bio-based polyethylene (PE), polypropylene (PP) or poly(ethylene terephthalate) (PET) (so-called “drop-ins”) and bio-based technical performance polymers such as poly(trimethylene terephthalate) (PTT) or thermoplastic copolyester elastomer (TPC-ET);bio-based and biodegradable material, such as poly(lactic acid) (PLA), poly(hydroxyalkanoate) (PHA) and poly(butylene succinate) (PBS);fossil-based but biodegradable material, such as poly(butylene adipate terephthalate) (PBAT).

**Figure 1 ijerph-19-13970-f001:**
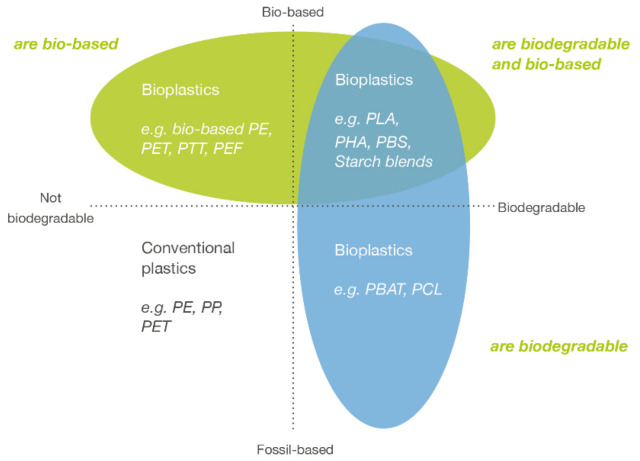
Material coordinate system of bioplastics [8].

As long as food and feed-like raw materials should be avoided for bioplastics production, there exists a strong need to seek alternative feedstock [9]. As stated above, one of the solutions might be to use wastes. However, the recent changes in the view of the economy as circular [3] renders the determination of some materials as wastes very complex and never straightforward [10]. Some materials called wastes in the past, e.g., organic fractions of municipal wastes, agriculture leftovers, etc., lost their attributes as they became relevant feedstocks for the energy and fine chemical sectors [11]. The food wastes bring another uncertainty in terms of definition. The food wastes are defined as the surplus of traded food that leads to its waste in the stages of retail sale, catering and households, which is understood as a reduction in the weight of food that was destined for consumption by humans [12]. The importance of the prevention of food losses and the creation of food waste is expressed in the “Farm to Fork Strategy” [13]. The European Commission states that the avoidance of the food loss and waste is key to achieving sustainability. Consumers, as the participants of the food chain production, are to play an important role in the implementation of this strategy. It is worth stressing that, in Europe, around 40% of food waste occurs at the consumer level [13]. The use of the intelligent and active food packaging [14] might be considered a useful tool to sort out this problem. Yet, the use of food wastes to deliver new food packaging materials requires a lot of effort in ensuring the consumer perception as well as legislative adaptations regarding the use of waste-derived plastics. Present-day food marketing focuses on the positive impact on health or well-being and on paying attention to the environmentally friendly approach and technology used to produce the food [15,16,17,18,19,20]. However, a ground-breaking approach to build up the social acceptance in food marketing is appealing to the idea of recycling food waste and bringing food waste back into the food loop. Another challenge is to highlight the environmental benefits to make them vital to the ordinary citizen, convincing her/him to accept bio-packaging for food at a higher price than cheaper traditional fossil counterparts. It is especially relevant as “sustainable” decision making is not always effective [21,22]. For example, unwrapping the food may be fueling, rather than combating, Europe’s food waste problem. [23]. According to this study, both packaging and waste doubled in the EU between 2004 and 2014. The solution does not seem to be to stop using packaging but rather to create better packaging characterized by lower environmental and social impacts.

Therefore, the main goal of this work is to answer the three main questions given in the title: Is there a global ban on plastic? What is the general social behavior in relation to the global exclusion of single-use plastics? Are consumers ready to replace plastic with the second-generation bio counterparts? This work addresses these challenges, especially in the context of better understanding to what extent consumers are guided in their everyday choices by concern regarding the environment and whether they would be willing to adopt new solutions based on waste in food packaging. 

## 2. Material and Methods

The online survey methodology was used to achieve the objectives of this study. The snowball sampling method for collecting data within the research was implemented. The snowball sampling method [24], or, the sampling method with a chain reference, is widely used where the group covered by the research is imprecise or difficult to specify, e.g., when the sampling group should be composed of all inhabitants of the earth, excluding only small children for whom parents make decisions, and the question of when to select the most reliable persons/objects as the reliable and representative research test group is problematic [25]. Simultaneously, defined as a probability-free sampling technique, this method takes samples, as indicated by the researchers. In the case of this work, recommendations for the recruitment of the questionees were provided by the authors of this work. In the first step, by indicating their own questionees, they were asked to forward the link to the questionnaires and the survey to other friends, thus creating a sampling with a chain reference [26].

This approach allowed for taking advantage of information and communication technologies, especially in terms of timesaving. In addition, in the COVID-19 pandemic time, an online survey allowed for the avoidance of direct contact between the questionees and interviewers. A potential disadvantage of the online survey is a risk of fraud by, e.g., multiple answers by the same questionee or using boots as questionees. Considering that there is no competing interest for any anonymous questionee, the risk of potential fraud was judged as minimal. Hence, contemplating these aspects, the survey was performed using the Edito CMS platform to conduct activities related to the content sharing. The survey questionnaire was created in the Dynamic Surveys module, which is a part of the Edito CMS platform. A questionnaire was designed as a single choice, and the questionee has an option to not respond to one or more questions.

The survey was made available on the RIC Pro-Akademia website and was prepared in English, Polish, Portuguese, Russian and Ukrainian by native speakers. The online survey was conducted worldwide between 16 June 2021 and 17 September 2021. 

The survey was divided into three parts. The first part was the introductory note, which is as follows: The survey is carried out by the Pro-Akademia Research and Innovation Center in Konstantynów Łódzki. Its results will be included in the project for the production of modern, biodegradable food packaging. We want to investigate whether consumers of bread, cakes and cookies follow the idea of environmental protection in their consumer choices and whether they would be willing to support innovative environmental solutions. Bioplastic can be the best ecological alternative to plastic, mainly produced from crude oil. It can be produced from plant raw materials or from waste biomass and therefore can be safer for users and the environment. 

The second part was focused on acquiring fundamental information about the socio-economic status of the questionee. The last part, composed of eight questions, was directly related to the use of bioplastics as food packaging material. The list of questions and possible answers is presented in Table 1.

The international authorship allowed for gathering the legal framework for the surveyed countries in the local language.

### Statistical Data Analysis

The survey data were analyzed using descriptive and inferential statistics. Both statistical approaches were implemented using Microsoft Excel software. The inferential statistics were used to test the significance of the hypothesis and the results to validate that the drawn conclusions reflect the reality on the ground, not random chance. Multiple regressions were used to examine the relationships. The statistical significance of regression coefficients and effects were assessed using analysis of variance (ANOVA), with a *p*-value < 0.05 considered statistically significant.

## 3. Results and Discussion

To better understand the consequences and potential of a global ban on plastic and the more extensive use of bioplastics, this work scrutinized the legal frameworks and analyzed the general social acceptance for alternatives of single-use plastics in food and beverages in the most representative surveyed countries.

### 3.1. Legal Framework

#### 3.1.1. European Union

In the European Union (EU), the key regulation in terms of plastics is the EU Directive 2019/904 on the reduction of the impact of certain plastic products on the environment [27]. This directive, also often called the Single-use Plastics (SUP) Directive, has introduced various measures for reducing the quantity of plastic goods being produced, along with measures for encouraging redesign and collection for recycling. However, it is important to realize that this is not the first approach to controlling the use of single-use plastics. The preceding directive was the Plastic Bags Directive (EU Directive 2015/720) [28] that was also an amendment to the Packaging and Packaging Waste Directive (94/62/EC) [29]. The last one was the first attempt to deal with the unsustainable consumption and use of lightweight plastic carrier bags (i.e., plastic carrier bags with a wall thickness below 50 microns), which, at that time, was already one of the top ten littered items in Europe [30]. Such lightweight plastic carrier bags are often used only once and, at the same time, they take up to 20 years to break down in the marine environment [31]. They often end up in terrestrial or marine animals’ digestion systems or break up into microplastics that ultimately go into the human and animal food chain [32]. Currently, the SUP Directive imposes many extended restrictions on the use of plastic for, e.g., cups for beverages, including their covers and lids; food containers intended for immediate consumption, either on-the-spot or take-away-ready for immediate consumption, etc. However, what is even more relevant is that the Single-Use Plastics Directive introduces a complete ban for a wide variety of plastic products, e.g., cotton bud sticks, cutlery (forks, knives, spoons, chopsticks), plates, straws, beverage stirrers, sticks to be attached to and to support balloons, food and beverage containers and cups made of expanded polystyrene with or without a cover, cap and lid. Additionally, the same directive puts enhanced responsibility on the producers. Producers are responsible for the introduction of certain types of plastics and should cover the costs that fall into one of the three categories, namely, the costs of the awareness-raising measures regarding those products; the costs of waste collection for those products that are discarded in public collection systems, including the infrastructure and its operation and the subsequent transport and treatment of that waste; and, finally, the costs of cleaning up the litter resulting from those products and the subsequent transport and treatment of that litter. In other words, the producers are forced to pay the potential environmental damages or seek more environmentally benign materials. 

Although the proposed changes are in line with the objectives of a more sustainable and less fossil-carbon-dependent future, numerous aspects still must be properly regulated. One of these is the life cycle assessment (LCA) of fossil- and bio-based feedstocks used for plastics production. For example, European Bioplastics [33] claims that the approach of the Joint Research Center of the European Commission (JRC) in the LCA study [34] lacks important elements that are crucial for a fair, comparative assessment of bio-based and fossil-based plastics. The proposed methodology’s approach of omitting the added value of biogenic carbon sequestration is a main criticism addressed by European Bioplastics, and, as a result, the proposed LCA approach clearly favors conventional plastics made from fossil resources. Furthermore, European Bioplastics complains that the indirect land use change rules [35] provide less strict requirements for fossil-based plastics. This results in an inconsistent inclusion of indirect effects, establishing different burdens of proof. The methodology also does not reflect the existence of multiple end-of-life realities and fails to treat all recycling options, including organic recycling, equally [36]. This single example depicts that a lot must still be done to properly address the changes in the plastic area in European and, later, in national legislative frameworks. 

Following their obligations, some EU member states already started the implementation of the SUP Directive to the national legal system. Such examples are Portugal and Poland.

#### 3.1.2. Portugal

In case of Portugal, the first stage of the SUP Directive transposition was to forbid the use and availability of single-use plastic tableware in the activities of the catering and/or beverage sectors and in the retail trade, which was to be in place starting on 2 September 2020 [37]. Due to the COVID-19 pandemic situation, this ban was postponed and entered into force on 31 March 2021. The second and the last part of the transposition was the entry in force, on 1 November 2021, of a new law. The objective of this law is to prohibit the placing of certain single-use plastic products on the market, such as those already mentioned for the EU legal framework [38]. At the same time, this new regulation sets two goals for reducing the consumption of cups for beverages and packaging for ready-to-eat foods by 80% by 31 December 2026 and by a 90% by 31 December 2030, compared to the values of 2022. To accomplish these goals, some measures are also planned to be implemented. One of them, in response to the need for the development of a circular economy and for addressing challenges related to the excessive use of single-use plastics during the COVID-19 pandemic crisis, is the creation of a system promoting the multiple use of food containers used for take-away or home delivery [39]. The implementation of this system is foreseen to be carried out either at the national scale or by individual food retailers or restaurants.

#### 3.1.3. Poland

In Poland, due to the necessity to implement the requirements of the SUP Directive, on 3 July 2021, the Polish Minister of Climate and Environment published a draft act amending the act on the obligations of entrepreneurs with regard to the management of certain types of waste and the product fee [40]. The new act aims to fully transcript the SUP directive in terms of a ban on single-use plastic products and the responsibilities of the entrepreneurs placing selected products on the market.

#### 3.1.4. Ukraine

During the 30th Ecology All-Ukrainian Forum on 7 June 2021, President Volodymyr Zelenskiy signed the Law on “Restricting the Trade in Plastic Bags in the Territory of Ukraine” No. 1489-IX, adopted by the Verkhovna Rada on 1 June 2021 [41]. The Plastic Act—the so-called “plastics policy” of the Ukrainian government—is aimed at ordering the issue of the use of plastics in Ukraine and identifying areas for improvement of the legislation, considering the specificity of domestic development related to the European approach to plastic waste management and the positive experiences of the other countries and leading companies as well. The Ukrainian government states that for the effective implementation of the Plastics Act in Ukraine, it is necessary to clarify the applicable national standards and harmonize them with European ones—first, with regard to the biodegradation and labeling of plastic bags. At the same time, attention is drawn to the fact that reducing the amount of plastic waste should go hand-in-hand with the search for substitutions for plastics, including alternative products and technologies, and changes in consumer habits. State legal regulations cover the issue of the reduction of plastic waste in several dimensions: (i) in terms of implementation—it is planned to create a state support mechanism for the replacement of the fossil plastics by the alternative environmentally friendly bioplastics, the implementation of technologies reducing the content of plastic in the final product and the effective enforcement of the principle of “polluter pays”; (ii) in terms of distribution, making retailers responsible for avoiding plastic in their distribution processes and (iii) in the field of recycling—the creation of a support system for plastics processing companies in order to implement the latest ecological recycling solutions into the Ukrainian economy.

#### 3.1.5. Brazil

In terms of the use of plastic materials and its potential recycling, in Brazil, no dedicated legal framework exists today. Plastic is considered as any other waste and falls under the solid residues law called the National Politics of Solid Residues that was established in 2010 [42]. Only now, following the examples of other countries, has Brazil considered the importance of creating a dedicated law regarding the use of disposable plastic [43].

#### 3.1.6. USA

In the last decade, regulations on the manufacturing and use of plastics, as well as subsidies to promote the recycling of plastic wastes, have been increasing in the USA [44]. Single-use plastics and carry-out bags have been banned in several American states, including Hawaii in 2011, California in 2014 and New York, Delaware, Oregon, Maine, Connecticut and Vermont in 2019. In addition, single-use straws and polystyrene have been restricted in Vermont since 2019, while the Hastings-on-Hudson city (at New York state) banned the use of polystyrene in 2014. Fees up to USD 1.00 have been imposed on carry-out bags, depending on their thickness. Interestingly, a solution was put in place in the state of Maine, where, since 1991, retail stores can make a plastic bag available to a customer as long as all plastic bags within 20 feet of the entrance are collected and then recycled [45]. Other American states, such as Arizona, North Dakota, Minnesota, Oklahoma, Iowa, Missouri, Michigan, Indiana, Mississippi and Florida, acted to prohibit local governments from banning plastic bags and containers [45]. It is important to acknowledge that, since the start of the COVID-19 pandemic, many US states such as Massachusetts, New York and New Hampshire have rolled back the policies restricting the use of single-use plastic bags [46,47].

#### 3.1.7. China

China has started to move away from the use of plastic straws and one-use plastic bags [48]. On 1 January 2021, a plastic prohibition went into effect in China, banning restaurants nationwide from using single-use plastic straws and shops in major cities from supplying plastic carrier bags. These are some of the restrictions on the production, sale and use of single-use plastic products set out in a policy on further strengthening the removal process of pollution from plastics issued jointly by the National Development and Reform Commission (NDRC) and the Ministry of Ecology and Environment on 16 January 2020 [49]. The NDRC obliges provinces across the country to implement, step by step, the national policy depicted within the five-year roadmap to reduce the use of single-use plastic products by the end of 2025. Non-degradable plastic bags will be banned in malls, supermarkets, pharmacies, bookstores and bars/eateries in all cities by the end of 2022. Only fresh produce markets are exempt from the ban until 2025. Restaurants across the country are already banned from using non-degradable plastic straws and plastic tableware. Take-away food packaging in cities must be cut by 30% by 2025. Hotels and private lodgings across China are forced to eliminate plastic items entirely by 2025 [50]. 

On top of that, plastic is banned in postal and courier services as well [51]. The specific developed areas of China, including Beijing, Shanghai, Jiangsu, Zhejiang, Fujian and Guangdong, are banned from using non-degradable plastic packaging until the end of 2022. This ban will be in force nationwide by 2025. Furthermore, according to the opinions of the NDRC, the production of cosmetics containing plastic microbeads is also excluded, and the sale of such products will be completely forbidden by the end of 2022 [49].

Under the Chinese act on solid waste, local authorities can impose a fine of CNY 10,000 to 100,000 (approximately USD 1545 to USD 15,460) for those who do not comply with the national restrictions on the use of non-degradable plastic bags and other single-use plastic products [52].

### 3.2. Consumer Acceptance for Alternatives to Single-Use Plastics in the Food and Beverage Sectors 

#### 3.2.1. European Union

The study performed by Pro Carton demonstrates that more than 60% of Europeans pay attention to the environmental impact of the packaging while purchasing a product [53]. Moreover, European customers are also influenced by the media coverage about the pollution, especially when it comes to marine pollution or microplastics that end up on consumers’ plates. Among alternatives to plastic, the most common are, e.g., glass and aluminum, especially for beverages, or paper-based materials for other uses. Another option, especially in the beverage industry, is the use of biodegradable plastics. Dominant producers in this market make significant efforts to demonstrate their commitment to using bottles made of renewable materials. These tendencies are consistent with changes in the social perception. During the COVID-19 pandemic, between 45% and 69% of Europeans, depending on the country, declared that they are recycling more than they did in 2020 [53]. These data demonstrate that the general perception of society is changing and that the declared acceptance of alternatives to single-use plastics in Europe is rising and is one of the major concerns of European society nowadays.

#### 3.2.2. Portugal

In Portugal, the average citizen produces 40.3 kg of plastic waste per year, which is 17% above the EU average [54]. The plastic packaging represents 8% of the overall Portuguese waste [54]. Only 13% of waste is separated for recycling and less than 20% is used for energetic valorization [55]. This means that the most predominant treatment in Portugal is landfilling (33%). This number can reach up to 58% when considering the reuse from other operations [56].

Between 2019 and 2020, the Portuguese recycling rate increased by 13%, which demonstrates an increase in awareness of the importance of recycling among the Portuguese population [56]. However, this is still not enough to mitigate the plastic problem in the country. Municipalities and companies have increased their recycling and reuse campaigns by creating drop-off containers for particular types of wastes such as coffee capsules, cooking oils, textiles and electronics, and, in some districts, PET bottles can be traded for price discounts [54]. Although these actions have led to an increase in recycling numbers, the trend is to enforce a reduction in the overall consumption by, for example, trading disposable capsules for long-term steel ones, hygiene liquid products for bars of soap, cleaning products for dissolvable tablets, etc. [55]. However, the main concern of plastic waste is food packaging. Although there is an increase in the sale of bulk food in supermarkets, reducing plastic packaging can lead to an increase in food waste, which, in Portugal, is already one-third of the overall food produced [57]. Therefore, bioplastics can play a major role in this reduction; however, bioplastics have also raised a few concerns, e.g., in the context of the environmental issues, especially regarding the insufficient information about the bioplastic reuse and correct separation approaches [58]. Nevertheless, Portugal is investing in bioplastics at both the research and implementation levels. For example, a consortium of 38 companies and investigation centers is planning to invest EUR 57.4 million in the construction of four factories, an R&D center and a logistics center for the production of insects and chitosan and the development of 43 insect-based materials and services, including bioplastics. At the same time, this investment will create 140 new direct jobs [59].

#### 3.2.3. Poland

Polish consumers have focused on recycling and reducing the use of plastic for many years [60]. The report prepared by Kantar, Europanel and GfK shows that, from year to year, Poles take more and more actions to ban plastic [61]. In addition, due to the COVID-19 pandemic, almost overnight, an overturned hierarchy of priorities among Polish consumers could be observed. The safe, quick and comfortable shopping became the key trend, but, despite the fear of being affected by SARS-CoV-2 and the tremendous number of plastic products created by the pandemic regime (e.g., disposable gloves, masks or one-use food packaging), the majority of Polish consumers declare that they have environmentally friendly attitudes, which have been created over recent years [4]. The percentage of Polish consumers who take specific measures to reduce plastic waste increased from 18% in 2019 to 21% in 2020 [62]. At the same time, the number of consumers who can be considered as a group disregarding the plastic pollution hazard as a serious environmental problem has decreased from 41% to 35%. Additionally, the plastic waste pollution remains the main global environmental problem for Poles. On their list of the greatest threats to the environment, the problem of the uncontrolled, catastrophic amount of plastic rubbish is in first place [62]. Occupying the following places on this list are climate change, air pollution and food waste. As many as 48% of consumers in Poland feel personally affected by environmental problems. Because of that, 31% of Poles have stopped buying certain one-use plastic products, 55% have used refill and reused packaging, if available, and 54% try to avoid plastic when buying fruits and vegetables. In addition, 83% of shoppers use their own shopping bag, and 77% skip plastic cutlery and plates for a party or barbecue. The biggest change in the attitudes of Poles took place regarding biodegradable cotton buds: even though the price of them is higher compared with that of the standard plastic buds, 34% of buyers declared to choose them, compared to 23% a year ago [62].

The Kantar’s “Who Cares? Who Does?” report from 2021 shows that 92% of Poles declared that they segregate plastic packaging always or often. To the question of who is most responsible for the elimination of single-use plastic packaging, 47% of Polish consumers answered that producers are, 29% answered that authorities and law regulators are, 4% answered that retailers and distributors are and 20% answered that it is society and consumers through their everyday purchasing decisions [61].

#### 3.2.4. Ukraine

Despite the extremely difficult political, economic and social situation, Ukrainian society is interested in environmental issues [63,64]. There is a public discussion on the “Ukrainian National Solid Waste Management Strategy”. More and more initiative groups and NGOs are being formed around the problems of environmental protection. Zero Waste Alliance Ukraine, an active member of the global environmentally friendly social movement, is represented by the Zero Waste Europe organization and was founded in 2018 by three Ukrainian non-governmental organizations—Zero Waste Lviv, Ozero (now the Zero Waste Society) from Kyiv and Kharkiv Zero Waste. Zero Waste Alliance Ukraine runs information and advocacy campaigns such as WeChooseReuse, EnvironMenstrual Week, Plastic-Free July, Brand Audit, etc. Nevertheless, there is still work to be done to raise awareness of the widespread social movement regarding the defense of the environment, as the country is still under continuous political and economic threats. 

#### 3.2.5. Brazil

Although there exists a general understanding of the importance of plastic pollution (Figure 2), there is no specific research on social acceptance for alternatives to single-use plastics in the food and beverage sectors. Most of the activities in this respect are related to the private sector. Private initiatives in the plastics sector and institutions have promoted better environmental awareness of this important material. For example, the aim of the Brazilian Association of Flexible Packaging Industry (ABIEF) is to develop plastic recycling programs and thus promote environmental education, seeking to recognize the plastic production chain as a value chain. Another institution, Plastivida, a social service organization, sees plastic as a relevant tool for sustainable development.

The Benchmarking Brazil Program, a respected sustainability seal that, since 2003, selects and certifies good practices of Brazilian organizations, catalogs approximately 400 cases that are considered benchmarking references for the quality of the practices adopted. Among them, the managerial themes of Productive Arrangements, Waste and R&D were those that registered innovative practices regarding renewable raw materials and waste management with social inclusion. Despite a large number of social movements regarding this subject, a lot must still be done to achieve a considerable impact at the national level.

#### 3.2.6. USA

In the USA, in addition to the increased generation of waste plastic bags, the relaxation of some policies, e.g., rolling back the policies restricting the use of single-use plastic bags due to the COVID-19, is likely to have long-term consequences on consumers’ attitudes towards re-introducing the single-use culture for consumers [46,47]. Improvements in the US plastic policies affecting the social awareness are strongly required, namely, in terms of popularity and transparency. It is clear that there is a lack of public awareness of plastic pollution; thus, the development of outreach and education throughout the population is quite important.

#### 3.2.7. China

Rapid economic growth and increasing prosperity in China are reflected in the rising problem of the amount of plastic waste that lingers in China. The social acceptance for littering everywhere, the absence of plastic recycling or reusing systems and the fact that China is the world’s biggest plastic producer and consumer takes the problem of plastic waste to the form of a national environmental disaster. When assessing social acceptance for pro-environmental activities in China, it is not possible to apply the same measures and directly compare with Western countries. These actions, including the ban on plastic, are generally initiated by the government, although grassroots, local social initiatives are also being observed [65,66,67]. In May 2018, the “Beautiful China Initiative” was announced, and it is a long-term environmental protection strategy aimed at fighting air, water and soil pollution and mitigating the effects of climate change, the depletion of resources and the habitats’ exhaustion [68]. The solid waste recycling industry in China has grown rapidly over the past five years [69]. The industry’s revenues have grown at a dizzying pace of 9.6% annually in the last 5 years, and for the current year, 2021, an increase of 10.2% compared to the previous year is expected. China’s plastic waste recycling market is worth USD 22.7 billion [69]. Despite these very positive signals, a lot still has to be done to change the social perception of this aspect.

### 3.3. Online Survey Results 

The online survey was conducted worldwide. The spread of answered surveys between countries is presented in Table 2. In total, 391 surveys were received from 16 different countries, while 9.5% were submitted with no country specified. Although the number of online surveys answered was low, the general trends discussed below can be drawn from the work of J. A. Angrist and G. W. Imbens, the Nobel prize winners in Economy for 2021, who demonstrated that the use of natural experiments in empirical social economics has ushered in the analysis of causal relationships [70]. Furthermore, the use of inferential statistics helped to extrapolate the drawn conclusion to the population size, i.e., the general audience.

By analyzing the number of questionnaires, it can be said that the representative countries were: EU countries—Poland and Portugal, China—the economy experiencing the fastest pace of development for the last decade [71], USA—the country with the highest level of Actual Individual Consumption (AIC) per capita in the world [72], Brazil—the country which faces many environmental problems of crucial importance for the entire world [73] and Ukraine, which is a characteristic example of a low-income post-communist country [74].

#### 3.3.1. Characteristics of Questionees

Regarding the sex and age of the questionees, 66.5% were women (*n* = 260). A total of 80.3% of questionees (*n* = 314) were between 21 and 65 years old, and only 6.4% (*n* = 25) were above 65 years of age. The age structure corresponds to the world population age structure [75] and demonstrates that the vast majority of the questionees were of the age of the actual active consumers. 

Taking into consideration the place of residence (Table 3), the majority (68.3%, *n* = 267) of questionees live in cities with a population of over 150,000 inhabitants. Taking into consideration the major differences in size and population between countries, it is hard to generalize regarding the representation of the questionees for each country individually. Nevertheless, from the survey point of view, it is important to stress that inhabitants of villages and towns constitute a considerable sample (above 17%, *n* = 68). This is in agreement with the worldwide structure of population, as currently 54% of people live in urban areas, and by 2050, the UN foresees that this number will increase to 68% [76]. Therefore, the results can be considered as representative of the public in the surveyed countries and in general.

Other relevant aspects in the analysis of the consumer preferences are the level of education and the economic status. The results are given in Figure 3. 

The analysis of the answers depicts that most answers came from questionees with secondary (17.4%, *n* = 68) or higher education (73.4%, *n* = 287). This share distribution was generally observed for all surveyed countries, with the exception of China, where secondary or higher education was claimed by 23.2% (*n* = 10) and 41.9% (*n* = 18), respectively. At the same time, primary education was claimed by more than ¼ of questionees (27.9%, *n* = 12), while overall, the share of questionees with a primary education was as low—5.4% (*n* = 21). 

In terms of economic status, there exists a general agreement between Ukraine, Portugal and Brazil. Between 49% and 58% of questionees declared the average economic status. In China, the dominant share of questionees (79.6%, *n* = 34) declared an average economic status, whereas, in Poland, the highest share of questionees (51.3%, *n* = 39) declared a good economic status. In general terms, an average economic status was declared by 50.6% (*n* = 198), followed by 31.2% (*n* = 122) of questionees who considered their economic status as good. An important conclusion to be drawn is that the questionees from different countries may consider the same economic status differently. For the ones from the USA, the country with the highest level of AIC per capita in the world, the average economic status may have a very different meaning compared to the meaning it has for the questionees from the low-income countries, as the comparison is not accompanied by any numerical indicator but only by the very subjective opinion of the questionee. Nevertheless, the explained differences, especially in terms of place of living, education and economic status, influence the profile of the typical questionee. However, overall, the average questionee was a woman (*n* = 260, SD = 91.3) between 21 and 65 years of age (*n* = 314, SD = 159.6) with higher education (*n* = 287, SD = 129.0) and an average economic status (*n* = 198, SD = 79.7) who lives in city with more than 500k inhabitants (*n* = 164, SD = 54.3). 

#### 3.3.2. Social Perception of Bioplastics 

This part of the survey consisted of eight questions, and the results are discussed below. 

Would you support replacing traditional plastic food packaging with bioplastic packaging?

The survey showed that there is a global consumer acceptance (91.2% positive answers, *n* = 301) for moving away from traditional plastic food packaging made from fossil fuels and replacing them with bioplastics. For all questionees, the problem of avoiding single-use plastic food packaging and generally limiting plastics in everyday life is important and necessary from a social and environmental point of view. The highest (100% (*n* = 7)) acceptance for the replacement of fossil plastics with their bio counterparts was noticed for answers from the USA. An almost equally high acceptance, 96% (*n* = 92), was found for Ukraine, (90% (*n* = 69)) Poland, (89% (*n* = 52)) Portugal, (88% (*n* = 45)) Brazil and (85% (*n* = 36)) China, as shown in Figure 4.

Although the collected results demonstrate a general agreement, it is worth verifying the hypothesis about a difference in social acceptance between questionees from European (M = 92.4%, SD = 3.9%, *n* = 3) and non-European countries (M = 90.6%, SD = 8.4%, *n* = 3) or between those from developed (Poland, Portugal, USA—M = 93.5%, SD = 5.7%, *n* = 3) and developing countries (Ukraine, Brazil, China—M = 89.6%, SD = 6.7%, *n = 3*). Neither the first hypothesis (, *t* (df = 4) = 0.33, *p* = 0.38 (1 tail)) nor the second one (*t* (df = 4) = 0.77, *p* = 0.24 (1 tail)) found its confirmation in the statistical analysis; hence, it can be concluded that there is indeed universal agreement on this matter for all questionees. Furthermore, these data are also coherent with those reported in the literature [77]. In addition, Scarpi et al. found that the widespread agreement on the change towards bioplastics can be associated with new moral norms and a new business-to-business (B2B) or business-to-consumer (B2C) relationship for circular economy success. 

2.Did you know that bioplastics can be made from leftovers of bakery and confectionery products (e.g., bread, cakes …)?

In the case of the first question, there was a generalized agreement between questionees; however, in the case of the second question, the responses were not fully conclusive. As much as 58.3% (*n* = 192) of questionees were not aware that bioplastics can be made from leftovers. Only Brazilian and Portuguese questionees were more aware about this, whereas, in all other countries, the dominant response was negative, reaching as high as 62.8% (*n* = 27) and 67.4% (*n* = 64) for China and Ukraine, respectively, as outlined in Figure 5. The t-test confirmed that the results obtained from the Chinese and Ukrainian questionees (M = 65.1%, SD = 3.3%, *n* = 2) are indeed statistically different (*t* (df = 4) = 2.84, *p* = 0.2 (1 tail)) from the answers of questionees from the remaining countries (M = 53.0%, SD = 5.3%, *n* = 4). Therefore, by analyzing the social structure of questions, it can be concluded that the noticeable difference can be associated with the fact that a considerable share of Chinese questionees have only primary education, and, in the case of Ukraine, there is still a low level of social awareness linked to an underdeveloped legal framework. Hence, it is important to verify if there is a relation between the level of education of questionees and a distribution of negative answers to question 2. For this purpose, the linear response surface model was used, which is described by the following equation:$$Y=\beta_{0}+\beta_{1}X1+\beta_{2}X_{2}+\beta_{3}X_{3}+\beta_{4}X_{4},\;$$
where *Y* is the percentage of negative answers to question 2, X_1_ to X_4_ are the shares of questionees with primary, vocational, secondary and higher education, respectively, and *β*s is the regression coefficients. $\beta_{0}$ represents the analyzed response in the center of the experimental domain. The statistical significance of regression coefficients and the effects were assessed using analysis of variance, and the results are given in Table 4.

The *p*-value of the adjusted model (*p* = 0.015) implied that the model was significant. Therefore, the statistical analysis demonstrated that the model was well adapted to the response (R^2^ = 0.999 and adjusted R^2^ = 0.995). All terms were the statistically significant model terms (*p* < 0.05). Regression coefficients were compared with each other to indicate the relative importance of each significant variable in the model equation. The regression coefficients showed that vocational education has an almost 10-fold higher negative effect than the primary education of questionees. The less influencing regression coefficient is the one associated with higher education, which is close to 100 times lower than *β*_2_. Therefore, the statistical analysis of the employed regression model confirms the hypothesis that there is a significant relation between the level of education and the perception that bioplastics can be made from leftovers of bakery and confectionery products.

3.Would it bother you to know that the bioplastic packaging, in which the bread is wrapped, is made of bakery and confectionery leftovers?

The analysis of answers to this question draws interesting conclusions. Although the vast majority of questionees (78.1%, *n* = 258) lack concerns regarding the new, waste-based bioplastics, there are some substantial differences between countries, as portrayed in Figure 6. The questionees from Portugal (100%, *n* = 58), Brazil (96.1%, *n* = 49) and Poland (85.5%, *n* = 65) are the most open-minded, and they accept the use of bakery and confectionary waste as a resource for a food packaging material production. The questionees from the USA (71.4%, *n* = 5) and Ukraine (69.5%, *n* = 66) were less willing to accept such a solution. For a great number of Chinese questionees (34.9%, *n* = 15), bioplastics made from leftovers raises doubts and opposition. Hence, it can be hypothesized that, in comparison to questionees from other countries (M = 93.9%, SD =7.5%, *n* = 3.), American, Ukrainian and Chinese (M = 58.6%, SD = 20.6%, *n* = 3) questionees demonstrate distinctive confidence regarding the safety and health issues of new products, combined with a low awareness about the technology advances. The inferential statistics allowed us to confirm this hypothesis (*t* (df = 4) = −2.79, *p* = 0.02 (1 tail)); thus, it can be concluded that the visions of questionees from the USA, Ukraine and China are indeed significantly different from those from the remaining countries.

Koenig-Lewis et al. [78] addressed this aspect from a different perspective. They examined the association of biopackaging with the type of food. Surprisingly, the biodegradable plastic used for unhealthy food did not affect the purchasing habits. The opposite effect was observed when healthy food was purchased. In this case, the biodegradable packaging was selected more often. Considering the results of Koening-Lewis and the results given in this work, it can be concluded that biopackaging might be a perfect solution for healthier foods, e.g., salad. Other studies confirmed a strong connection between the health consciousness and the impact of the purchase intention of food in compostable packaging [79,80].

4.Would you expect a clear notice on the bioplastic packaging that will allow you to choose between traditional plastic and biodegradable plastic?

By analyzing results of question 4, it can be concluded that the concerns about, e.g., the safety and health issues might in fact be the actual reason for the rather low acceptance of new bioplastics as a packaging material. Chinese questionees, followed by American and Ukrainian questionees, want to be fully aware of what they are buying, as can be observed in Figure 7. The observed results may suggest that this question is directly linked to the previous one. Therefore, again, it is relevant to verify to what extent the questionees from these three nations have a distinctive confidence regarding the bioplastic products. In fact, the statistical analysis confirmed the hypothesis (*t* (df = 4) = 2.86, *p* = 0.02 (1 tail)) that Chinese, American and Ukrainian (M = 87.7%, SD = 14.7%, *n* = 3) questionees have a distinct perception and wish to be clearly informed about the bioplastic products compared to the majority of Polish, Portuguese and Brazilian (M = 78.7%, SD = 4.1%, *n* = 3) questionees.

A clear and unambiguous notice about the product origin is the responsibility of companies/brands and governments. Therefore, this question can be seen in a broader context, e.g., to what extent governments and companies can do more to help the environment. Pro Carton, in their survey, showed that both the industrial sector and administration can be much more involved in and responsible for informing and promoting environmentally friendly packaging. According to their studies, an overwhelming majority of Europeans (76%) strongly agreed/agreed with this aspect. Furthermore, as many as 61% of Europeans approved the introduction of additional taxes to force brands and retailers to adopt more environmentally oriented norms and behaviors [53,81]. 

5.How much more would you be able to pay for a product packed in bioplastic?

An interesting conclusion, especially for producers and vendors, comes from the answers to question 5. Although the most obvious answer is the one indicating the lowest acceptable premium price, numerous studies on consumer declarations vs. purchasing decisions show that declarations do not always translate into purchasing decisions [78,82,83,84]. Hence, this question allowed us to compare the declaration of the awareness of environmental risks and the readiness to replace traditional plastic food packaging with bioplastic (answers to the question 1) with the levels of consumer consciousness, sensibility regarding environmental aspects and mental openness for changes. 

In general, the vast majority of questionees (*n* = 240) were ready to pay only up to a 10% premium price for new biopackaging, as shown in Figure 8. In all countries, the willingness to pay 10% extra for this kind of material is rather similar, with the exception of Brazil and Ukraine, whose respondents demonstrated that they are not able to pay extra fees. The responses from Brazilian questionees were more uniformly distributed among answers, i.e., 46% (*n* = 23) of questionees were able to pay up to 10% more, 20% of questionees (*n* = 10) were able to pay between 10% and 20% more, 4% of questionees (*n* = 2) were able to pay between 20% and 30% more and 8% of them (*n* = 4) were able to pay between 30% and 40% more than they do for traditional plastics. 

As stated above, Brazilian and Ukrainian questionees (22%, *n* = 11 and *n* = 21, respectively) were the least willing to be charged additionally for bioplastics. The answers of the Brazilians are really puzzling considering that almost 1/3 of Brazilians (*n* = 16) were able to pay more than 10% for bioplastics. This shows that the questionees representing Brazilian society were very distinct. On the other hand, similar to Ukraine, many households in Brazil are low-income, even if they self-declare as having an average or good economic status, as claimed in this work. The statistical analysis allowed for the confirmation (*t* (df = 4) = 2.41, *p* = 0.04 (1 tail)) of the veracity of the hypothesis that, in comparison to other nations (M = 9.8%, SD = 6.8%, *n* = 4), Brazilians and Ukrainians (M = 22.1%, SD = 0.1%, *n* = 2) have significantly different shares of the population who are not willing to pay any extra cost. So, it can be hypothesized that, to some extent, the economic status of each questionee defines a potential aptitude to pay or not pay the extra cost for bioplastic. This hypothesis was examined, and, for this purpose, the linear response surface model was used, which is described by the following equation:$$Y=\beta_{0}+\beta_{1}X1+\beta_{2}X_{2}+\beta_{3}X_{3}+\beta_{4}X_{4},\;$$
where *Y* is the share of those not willing to pay any extra cost, X_1_ to X_4_ are the shares of questionees with very good, good, average and bad self-declared economic status, respectively, and *β*s is the regression coefficients. $\beta_{0}$ represents the analyzed response in the center of the experimental domain. The statistical significance of regression coefficients and the effects were assessed using analysis of variance, and results are given in Table 5.

The *p*-value of the adjusted model (*p* = 0.009) implied that the model was significant. Therefore, the statistical analysis demonstrated that the model was well adapted to the response (R^2^ = 1.000 and adjusted R^2^ = 0.999). All terms were statistically significant model terms (*p* < 0.05). Regression coefficients were compared with each other to indicate the relative importance of each significant variable in the model equation. The regression coefficients showed that a self-declared bad economic status indeed has the highest positive effect towards not paying any extra cost. The regression coefficient representing the questionees with a very good economic status validated the fact that they are strongly willing to pay (the second highest regression coefficient with a positive value). The regression coefficients representing questionees with good and average economic statuses are similar in terms of coefficient value; however, apparently, the *β*_2_ is positive, i.e., questionees with a good economic status are less willing to pay any additional cost for bioplastics. Regardless of this, the statistical analysis of the employed regression model confirms the hypothesis that there is a clear relation between the self-declared economic status and the share of questionees with no ability to pay an extra cost for the use of bioplastics instead of fossil counterparts.

Similar conclusions can be drawn from the studies conducted by Pro Carton. This survey showed that society is only willing to pay the lowest possible amount of extra costs for packaging material, and only if this would mean a lower environmental impact of the final product [53]. As many as 61% of questionees declared that they were able to pay up to 10% more, and this share is, in general, consistent for all age groups included in this study. 

6.Did you know that bioplastic can extend the life of a product and preserve its visual and taste properties?

The collected responses to question 6 indicate that a great proportion of questionees have almost no knowledge about the potential positive impact of bioplastic food packaging on the food packed in it. The highest knowledge of the functionality of bioplastic food packaging in extending the life of a food product and preserving its visual and taste properties was observed in the answers from Portugal (44.8%, *n* = 26). The Chinese (39.5%, *n* = 17), Ukrainian (26.3%, *n* = 25) and Brazilian (23.5%, *n* = 13) questionees were aware of the positive values of bioplastic, but the Poles (19.7%, *n* = 15) and Americans (14.3%, *n* = 1) apparently know the least about it (Figure 9). Hence, it can be hypothesized that Portuguese, Chinese, Ukrainian and Brazilian (M = 34.0%, SD = 9.6%, *n* = 4) questionees demonstrate different degrees of knowledge compared to Poles and Americans (M = 17.0%, SD = 3.8%, *n* = 4). Indeed, the statistical analysis results (*t* (df = 4) = −2.29, *p* = 0.04 (1 tail)) confirmed that there is a significant difference in the understanding of this subject. Independently of this, the rather low understanding of this issue confirms that there is a lot to be done in terms of the better promotion of bioplastics not only as a solution for environmental problems but also as a solution for the reduction of food wastes and for better-quality food products.

7.Would you share the knowledge about such packaging with others?

As many as 86.7% (*n* = 286) of questionees declared that they were ambassadors of the biodegradable food packaging material. The Portuguese (96.5%, *n* = 56) turned out to be the most involved in sharing the new knowledge, while the Ukrainians (78.0%, *n* = 74) and Americans (71.4%, *n* = 5) seem to have the lowest willingness to transfer the information to others, as shown in Figure 10. Although the differences between Ukrainians and Americans (M = 74.7%, SD = 4.6%, *n* = 2) and the questionees from the remaining countries (M = 90.8%, SD = 4.6%, *n* = 4) are statistically significant (*t* (df = 4) = −4.0, *p* = 0.01 (1 tail)), they are still small and can be generalized as positive feedback. On the other hand, it can be seen as overoptimistic, as it is only related to a declaration, with no actual action involved from the questionees’ side. Nevertheless, it can be seen as a positive sign of social awareness in this context.

8.Would you encourage others to use modern, environmentally friendly packaging?

Similar to the previous question, 89.1% (*n* = 294) of all questionees declared green self-identity and stated that they would promote the use of modern friendly packaging. These results are rather puzzling considering the previous questions related to the potential use of bioplastics for food wrapping as well as paying a potential extra cost for bioplastics, where the answers were quite modest compared to those to this question. It can be hypothesized that the questionees are not consistent in answering questions about similar matters. The last question received a very positive response: 100% (*n* = 7) of American questionees, 98.3% (*n* = 57) of Portuguese questionees, 96.1% (*n* = 49) of Brazilian questionees, 88.2% (*n* = 67) of Polish questionees, 86.0% (*n* = 37) of Chinese questionees and 81.1% (*n* = 77) of Ukrainian questionees said that they would promote this biomaterial in their circle of friends and encourage others to use it, as shown in Figure 11.

The last two questions, although strongly declarative from the questionee side, demonstrate that society is aware of the plastic problems. Similar conclusions were found by Scarpi et al., who reported that the increased awareness of sustainability issues has increased the number of people following lifestyles oriented toward sustainable consumption [77]. This indicates that, behind the reported commitment to switch towards bioplastics and to promote their use, there exists a real change in the social perception. 

## 4. Conclusions

This study is not meant to be conclusive. Rather, several limitations illuminate useful directions for future research and further implementation. The replacement of traditional single-use plastics with bioplastics for food packaging produced using the second-generation resources is a long-lasting process of creation. The specific, semi-qualitative research method—the snowball sample applied to a group of 391 questionees from 14 countries in the virtual space—allowed us to collect interesting conclusions and make the first comparative analyses. The sample size was not strictly defined, as the questionnaire was disseminated on the internet. In all the countries covered by the study, work on the construction of the basic frameworks is underway. It considers (i) the legal framework, including tax and incentive systems for producers and consumers, (ii) technologies for producing environmentally friendly, biodegradable materials with characteristics and functionalities comparable to or even better than traditional plastic and (iii) global social acceptance and consumer acceptance for pro-environmental activities and products. However, as the survey conducted in Brazil, China, Poland, Portugal, Ukraine and the USA showed, consumers’ knowledge about the possibility of using the second-generation feedstocks, such as waste from bakeries or confectionery production, is considerably low. On the other hand, the consumers in the surveyed countries do not mind that the material produced from waste comes into contact with food, although some doubts in this respect are expressed, predominantly by Chinese questionees, followed by the ones from the USA and Ukraine. Most questionees in almost all surveyed countries are willing to pay only up to 10% more for new packaging. However, as many as 22% of questionees from Ukraine do not want to pay extra for biodegradable packaging at all. It is interesting that the questionees from China understand better than the questionees from other countries and accept that the bioplastic packaging must cost more; therefore, they are able to pay a higher price for this. In summary, this study shows that most consumers in all surveyed countries are ready to share the knowledge about new bioplastic food packaging materials and encourage others to move away from traditional plastic ones.

## Figures and Tables

**Figure 2 ijerph-19-13970-f002:**
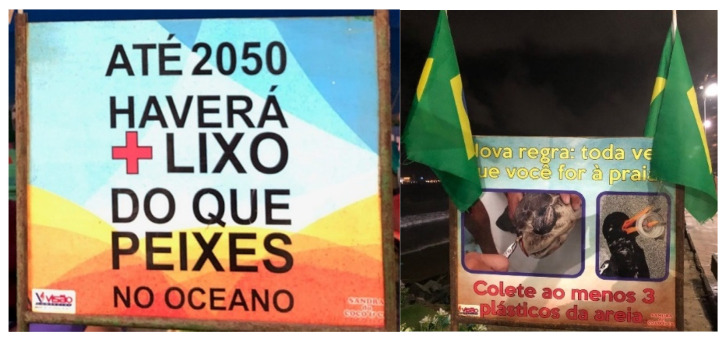
The banners at the Fortaleza (Ceará, Brazil) beach, made by aware citizens. Left photo: “By 2050, there will be + waste in the ocean than fishes”; Right photo: “New rule: every time you go to the beach, collect at least 3 plastics from the sand”. Photos taken on 6 March 2019 by a co-author of this work, R. M. Lukasik.

**Figure 3 ijerph-19-13970-f003:**
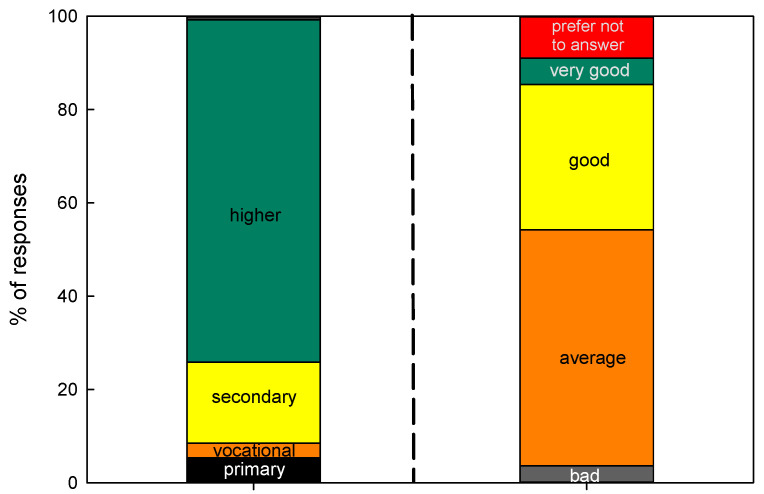
The level of education (**left bar**) and economic status (**right bar**) structure of questionees.

**Figure 4 ijerph-19-13970-f004:**
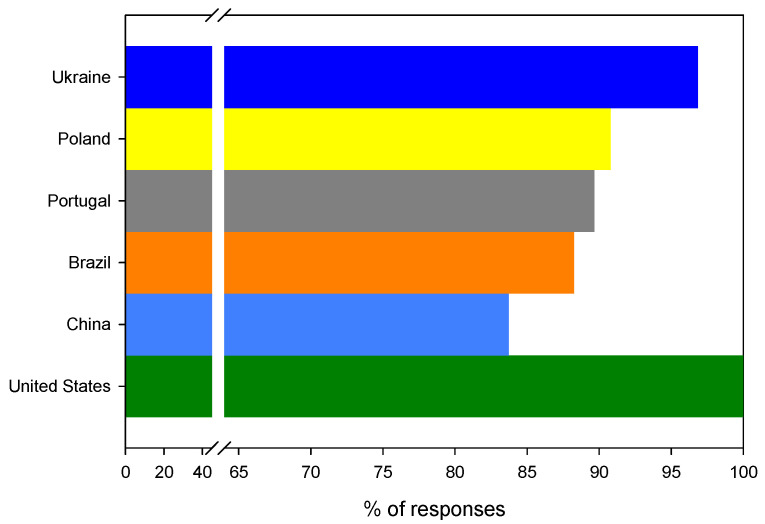
The distribution of positive answers to the question: Would you support replacing traditional plastic food packaging with bioplastic packaging?

**Figure 5 ijerph-19-13970-f005:**
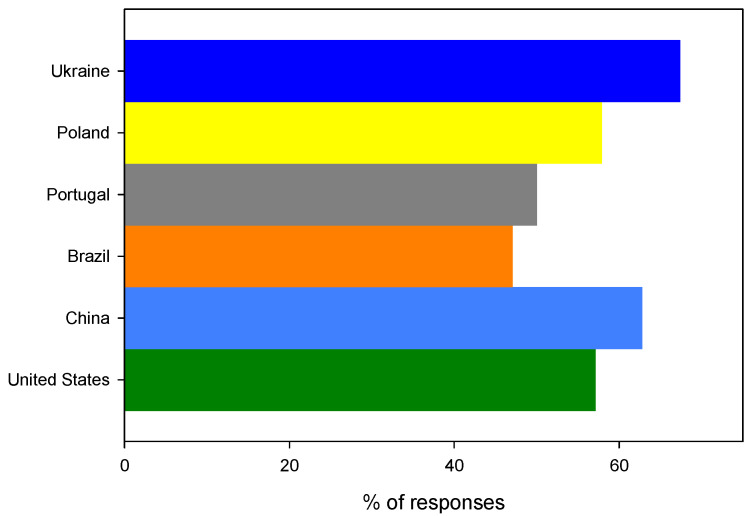
The distribution of negative answers to the question: Did you know that bioplastics can be made from leftovers of bakery and confectionery products (e.g., bread, cakes …)?

**Figure 6 ijerph-19-13970-f006:**
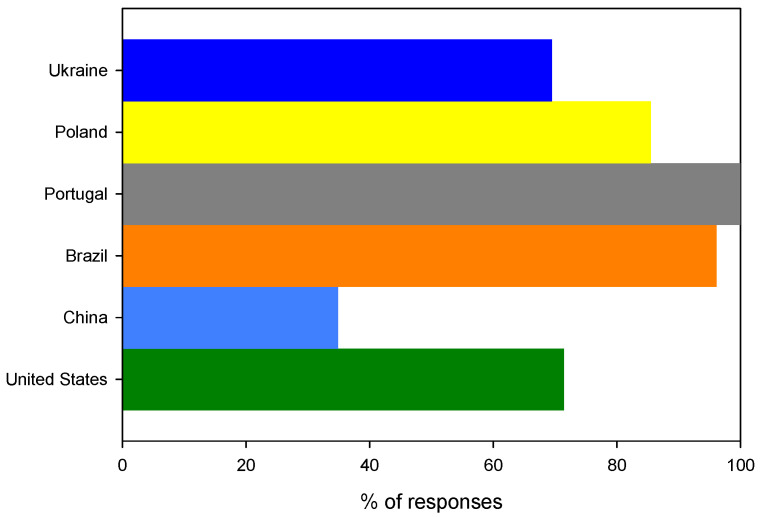
The distribution of positive answers to the question: Would it bother you to know that the bioplastic packaging, in which the bread is wrapped, is made of bakery and confectionery leftovers?

**Figure 7 ijerph-19-13970-f007:**
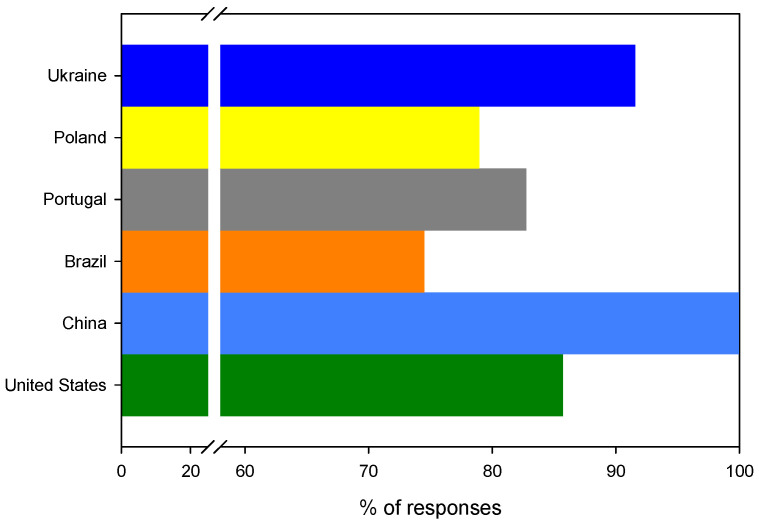
The distribution of positive answers to the question: Would you expect a clear notice on the bioplastic packaging that will allow you to choose between traditional plastic and biodegradable plastic?

**Figure 8 ijerph-19-13970-f008:**
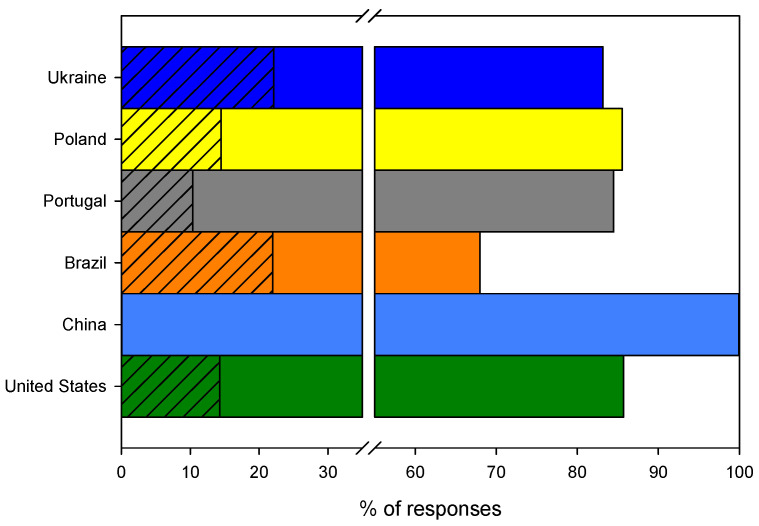
The distribution of answers of “not able to pay extra” (coarse bar) and “below 10%” (full bar) for the question: How much more would you be able to pay for a product packed in bioplastic? The data are given in a stacked form.

**Figure 9 ijerph-19-13970-f009:**
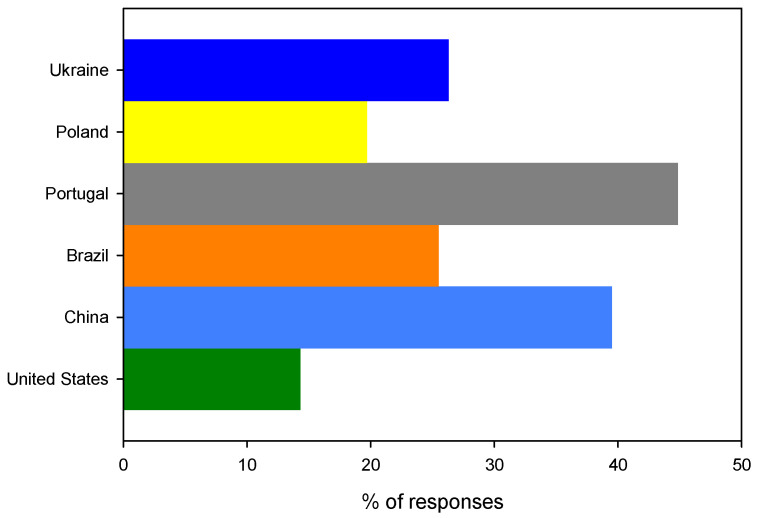
The distribution of positive answers to the question: Did you know that bioplastic can extend the life of a product and preserve its visual and taste properties?

**Figure 10 ijerph-19-13970-f010:**
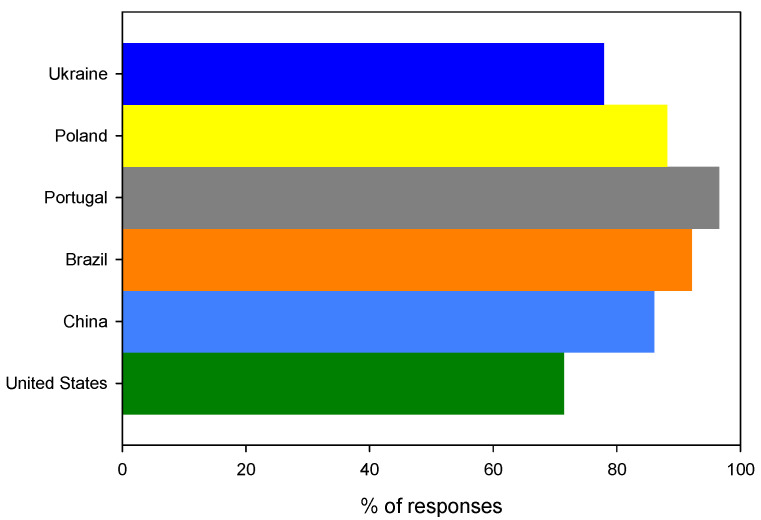
The distribution of positive answers to the question: Would you share the knowledge about such packaging with others?

**Figure 11 ijerph-19-13970-f011:**
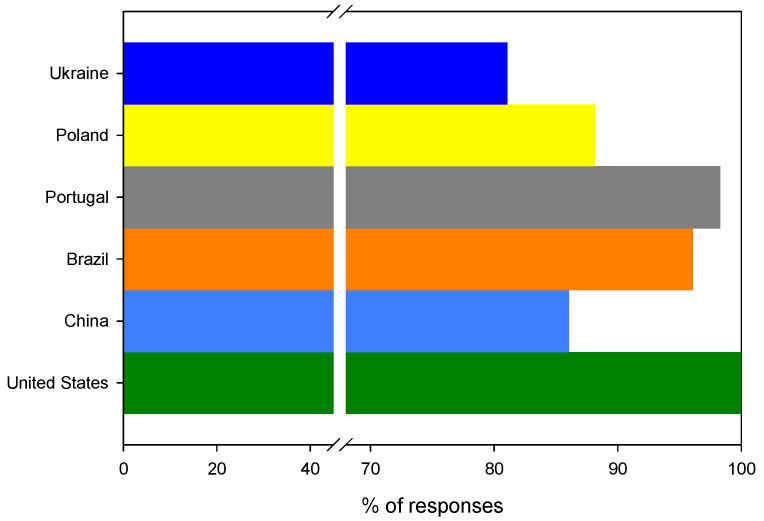
The distribution of positive answers to the question: Would you encourage others to use modern, environmentally friendly packaging?

**Table 1 ijerph-19-13970-t001:** The list of questions and available answers of the online survey.

Question	Answer
Gender	Female
	Male
	Other
	Prefer not to answer
Age	Under 21 years old
	21–64 years old
	Over 65 years old
Level of education	Primary
	Vocational
	Secondary
	Higher
	No answer
Economic status	Very good
	Good
	Average
	Bad
	Prefer not to answer
Place of residence	Village
	Town with up to 50,000 inhabitants
	City with 50,000 to 150,000 inhabitants
	City with 150,000 to 500,000 inhabitants
	City with more than 500,000 inhabitants
Who in your household makes purchasing decisions	Mostly me
	Me and another person
	Mostly another person
Are you interested in receiving survey results? If so, please provide your e-mail address.	Free text
Country of the questionee	Free text
1. Would you support replacing traditional plastic food packaging with bioplastic packaging?	Yes
	No
	I have no opinion
2. Did you know that bioplastics can be made from leftover of bakery and confectionery products (e.g., bread, cakes …)?	Yes
	No
3. Would it bother you to know that the bioplastic packaging, in which the bread is wrapped, is made of bakery and confectionery leftovers?	Yes
	No
	I have no opinion
4. Would you expect a clear notice on the bioplastic packaging that will allow you to choose between traditional plastic and biodegradable plastic?	Yes
	No
	I have no opinion
5. How much more would you be able to pay for a product packed in bioplastic?	Not able to extra cost
	Below 10%
	Between 10% and 20%
	Between 20% and 30%
	Between 30% and 40%
	More than 40%
6. Did you know that bioplastic can extend the life of a product and preserve its visual and taste properties?	Yes
	No
7. Would you share the knowledge about such packaging with others?	Yes
	No
	I have no opinion
8. Would you encourage others to use modern, environmentally friendly packaging?	Yes
	No
	I have no opinion

**Table 2 ijerph-19-13970-t002:** The number of questionnaires by countries of questionees.

Country	Questionnaires
Ukraine	95
Poland	76
Portugal	58
Brazil	51
China	43
USA	7
Cyprus	5
Greece	5
Mexico	5
Germany	2
Russia	2
Finland	1
The Netherlands	1
Ireland	1
Italy	1
Spain	1
No country indicated	37

**Table 3 ijerph-19-13970-t003:** The place of residence of questionees.

Place of Residence	Number of Inhabitants	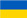	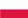	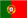	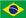	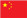	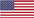	OtherCountries	No Country Indicated	Total
village		3	14	3	1	2	0	1	6	30
town	<50 k	5	5	13	4	4	1	3	3	38
city	50 k–150 k	4	15	12	6	10	1	6	2	56
150 k–500 k	42	2	7	13	8	3	2	8	85
>500 k	41	40	23	27	19	2	12	18	182

**Table 4 ijerph-19-13970-t004:** ANOVA statistical analysis for the response surface linear model of the % of negative answers to the question “Did you know that bioplastics can be made from leftovers of bakery and confectionery products (e.g., bread, cakes …)?” in the function of the education of questionees.

	Model	*β_1_*	*β_2_*	*β_3_*	*Β_4_*	R^2^	Adjusted R^2^
Coefficient		221.98	−2049.61	129.74	22.03	0.999	0.995
*F* value	2472.61	995.72	6238.98	1240.82	242.18		
*p*-value	0.015	0.020	0.008	0.018	0.041		

**Table 5 ijerph-19-13970-t005:** ANOVA statistical analysis for the response surface linear model of the share of questionees not willing to pay any extra cost for bioplastic implementation in the function of the self-declared economic status of questionees.

	Model	*β_1_*	*β_2_*	*β_3_*	*β_4_*	R^2^	Adjusted R^2^
Coefficient		−503.73	195.79	−150.97	706.62	1.000	0.999
*F* value	6327.97	3044.93	8207.04	11,527.27	16,469.54		
*p*-value	0.009	0.012	0.007	0.006	0.005		

## Data Availability

The datasets used and analyzed during the current study are available from the corresponding author on reasonable request.

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
