# Peer review of "Global Ban on Plastic and What Next? Are Consumers Ready to Replace Plastic with the Second-Generation Bioplastic? Results of the Snowball Sample Consumer Research in China, Western and Eastern Europe, North America and Brazil"

_ijerph, 2022, doi:10.3390/ijerph192113970_

Round 1

Reviewer 1 Report (New Reviewer)

Kochanska et al worked on the comparative study of the conventional plastics and bioplastics used as food packaging. In their research, they conducted online surveys iinvestigationent countries and concluded their research based on their questionnaire. 

As per my understanding, thecostshors should incorporate mbeforering cost and additional benefits prior to online surveys so that the audience should also have a broader perspective of the idea that they asked for. However, I would still recommend this study to be published with some changes. My suggestions are givewouldoprefer1.     Title is very long,clearly representa shorter title that may clearly epresent the work

1.     On page number 3, the statement such as: “Present-day food marketing focuses on the positive impact on health or well-being and on paying attention on the environmentally friendly approach and technology used to produce the food. ..”  you may cite the following aticles:

  doi.org/10.3390/ma15031046

doi: 10.3390/membranes12070701

doi.org/10.1016/j.jclepro.2020.124265

doi.org/10.1016/j.crfs.2021.07.005

3.     Would be a very nice and valuable addition if the manufacturing cost comparison and lifetime of conventional plastics and bioplastics is discussed in a separate paragraph.

4.     Since not all food products can be packaged in bioplastics, would be very interesting if the authors classify various products that are suitable for the bioplastics packaging and then built up a story on different questions.

5.     Text in the reference section seems different from the main body of the manuscript, please use a consistent format throughout the document.

In the first line of conclusion section, I don’t think there should be a period after the word conclusive.

Author Response

Reviewer 2 Report (New Reviewer)

Interesting and important topic - the consumer has significant impact on the use of biobased and/or biodegradable plastics. It is interesting to see the difference of consumer attitude and knowledge in different markets.

A few notes, slightly difficult to address as there are no line numbers:

p2. '..and the European Commission have..' - add reference

p3. '..a lot of efforts in ensuring the consumer perception.' - maybe also mention legislative factors affecting waste-derived plastics

p3. 'Nearly a decade ago, a UK supermarket experimented with removing all fruit and vegetables from the packaging – and the rate of such kind of food wastage has doubled.' - add reference

p6. 3.1. Legal framework - should this be a part of Introduction?

p7. 'single-use plastic products, such as cotton swabs, cutlery, plates, straws and balloon sticks' - this is repetition to 3.1.1

Figures 4, 5, 6, 7, 9 ,10, and 11 are messed up in the pdf -fix.

p21. Conclusions - Could the results be biased? E.g., the majority of responders were highly educated women. Was the selected approach to contact potential responders the best or if it was made differently, would the findings be different? Please discuss.

Author Response

This manuscript is a resubmission of an earlier submission. The following is a list of the peer review reports and author responses from that submission.

Round 1

Reviewer 1 Report

Dear Authors,

I found the paper relevant, but I think there are quite some issues that needs further consideration:

Title: the use of the term snowball sampling is rather confusing.

Snowball sampling is a recognized and viable method of recruiting study participants not easily accessible or known to the researcher. Usually the researcher does not directly recruit participants but contacts others who then connect them to research participants. Please explain how this process was organized? Particularly when considering the diversity of countries and cultures.

Material and methods

My major concerns are the sampling technique and sample size. I am not truly convinced the sampling technique could be considered as snowballing. The sample size is very low to draw any conclusion and provide meaningful insights. The data analysis does not go beyond descriptive statistics that is much less what expected from scientific paper.

The questionnaire is composed of a set of very basic questions. Some of them e.g. Would you support replacing traditional plastic food pack-aging with bioplastic packaging should not be asked without prior introduction of the whole concept of bio packaging. It is very risky to make the assumption that consumer can understand and correctly interpret what is behind.

The categories in the Q5 are not well defined. Below 40%? 10%, 20%, 30% is also below 40.

The questions like “Did you know that bioplastic can extend the life of a product and preserve its visual and taste properties?” can bias the results.

The results do not provide much meaningful insights as the sample is very diverse in terms of cultural background and the questionnaire and data analysis do not adequately tackle that.

The discussion of main findings is missing and the conclusions refers to comparative analysis but do not include any final thoughts on that.

Author Response

Dear Reviewer,

Thanks a lot for your valuable comments to our article. Please find our point-by-point answers enclosed.

Ewa Kochanska

Reviewer 2 Report

The authors seem to do some corrections. This is a resubmit paper but I can not find the point-by-point response and the revised areas of the manuscript. Please provide those information to me for review if it is convenient.

Author Response

Dear Reviewer,

Thanks a lot for your valuable comments to our article. Please find below our point-by-point responses.

Ewa Kochanska

Round 2

Reviewer 1 Report

Dear Authors,

the paper brings forward issues of high relevance, but my major concern is its cross cultural dimension as the sample size is very small.

with best regards,